# Influence of Cu^2+^ Ions on the Corrosion Resistance of AZ31 Magnesium Alloy with Microarc Oxidation

**DOI:** 10.3390/ma13112647

**Published:** 2020-06-10

**Authors:** Madiha Ahmed, Yuming Qi, Longlong Zhang, Yanxia Yang, Asim Abas, Jun Liang, Baocheng Cao

**Affiliations:** 1School of Stomatology, Lanzhou University, Lanzhou 730000, China; madiha73837719@yahoo.com (M.A.); zhanglonglong13@lzu.edu.cn (L.Z.); yangyx18@lzu.edu.cn (Y.Y.); 2State Key Laboratory of Solid Lubrication, Lanzhou Institute of Chemical Physic, Chinese Academy of Sciences, Lanzhou 730000, China; qym87@sina.com; 3School of Physics Science and Technology, Lanzhou University, Lanzhou 730000, China; asim.karam1985@hotmail.com

**Keywords:** AZ31 magnesium alloys, microarc oxidation, Cu^2+^-containing coating, corrosion resistance, cytocompatibility

## Abstract

The objectives of this study were to reduce the corrosion rate and increase the cytocompatibility of AZ31 Mg alloy. Two coatings were considered. One coating contained MgO (MAO/AZ31). The other coating contained Cu^2+^ (Cu/MAO/AZ31), and it was produced on the AZ31 Mg alloy via microarc oxidation (MAO). Coating characterization was conducted using a set of methods, including scanning electron microscopy, energy-dispersive spectrometry, X-ray photoelectron spectroscopy, and X-ray diffraction. Corrosion properties were investigated through an electrochemical test, and a H_2_ evolution measurement. The AZ31 Mg alloy with the Cu^2+^-containing coating showed an improved and more stable corrosion resistance compared with the MgO-containing coating and AZ31 Mg alloy specimen. Cell morphology observation and cytotoxicity test via Cell Counting Kit-8 assay showed that the Cu^2+^-containing coating enhanced the proliferation of L-929 cells and did not induce a toxic effect, thus resulting in excellent cytocompatibility and biological activity. In summary, adding Cu ions to MAO coating improved the corrosion resistance and cytocompatibility of the coating.

## 1. Introduction

One of the most crucial topics in the biomaterial field is the development of degradable biomaterials [1,2,3,4]. Metallic biomaterials are widely used in dentistry, orthopedics, and cardiovascular medicine [4,5]. Given their biological stability and excellent mechanical and processing properties, metallic biomaterials play an important role in implant applications [6,7,8]. Implants are used to reconstruct a failed tissue; therefore, a traditional biomaterial frequently requires a second surgery for removal [2]. The skeletal anchorage in orthodontics, such as miniscrews and miniplates made of magnesium alloys, provides stable implant materials that degrade in vivo [9,10], eliminating the need for a second operation for implant removal and helping to overcome the limitations of conventional orthodontic techniques [11].

The AZ31 Mg alloy consists of Al (2.5–3.5%), Zn (0.6–1.4%), Mn (0.2–1.0%), and balance Mg [12]. Al and Zn enhance and accelerate the hardening, while Mn increases the corrosion resistance [2]. The greatest advantages for Mg alloys are their fracture toughness and elastic modulus, which have a greater resemblance to those of bone when compared with other metallic implants [2,13,14]. Mg is a common element in the body that is essential for regulating metabolism [2]. Mg alloys are nontoxic materials [2]. Nevertheless, they have substantial limitations. For example, they corrode rapidly in body fluid. The rapid corrosion could form H_2_ bubbles in the injured tissue, which delays and affects the healing process. Moreover, the alkaline pH around the corroded area will change [2,15]. The fast absorption and accumulation in the body will block the bloodstream and may cause death for the patient [2,15].

To overcome these shortcomings, enhancing the corrosion resistance and cytocompatibility of the AZ31 Mg alloy coating is essential for providing a low-solubility barrier that separates Mg alloys from the environment. Microarc oxidation (MAO), better known as plasma electrolytic oxidation, is the most used among all surface modification technologies that are available for coating [2]. An oxide layer can be formed using the MAO method, which increases the corrosion resistance of the AZ31 Mg alloy and exhibits a better wear resistance than other methods [2,16]. Moreover, in MAO coatings, cracks and pores will form, providing the coating on a substrate with high bond strength [2]. These pore surfaces that are formed can enhance cell proliferation/adhesion, which leads to rapid healing of local tissues. A MAO-coated layer also has a low toxicity property; thus, this method is used for medical tools and devices, such as screws, pins, and plates [10,17,18,19]. Many recent studies have explored the properties of Cu^2+^, considering that it has an antibacterial property [20,21,22]. Cu^2+^ is an essential element for living organisms [22,23]. It is a vital trace element that plays a key role in immunity, increasing the average rate of bone resorption in bone metabolism and improving collagen fiber precipitation [20,24,25]. It is also important in the areas of bone engineering and regeneration [20].

A Cu^2+^-containing coating provides a better protective surface against corrosion compared with other coatings [26,27,28,29,30]. Wu et al. demonstrated that using the MAO method to apply a Cu–TiO_2_ coating on Ti showed an excellent antibacterial activity and a good corrosion resistance [22]. Zaki et al. [31], Kamil et al. [32], and Van Phuong et al. [30] enhanced and improved corrosion resistance by experimenting with Cu ion coatings. Cu ions in an electrolyte make a coating thick and decrease its porosity, leading to an improved corrosion resistance. In biomedical uses, the amount and tolerance of supplementary Cu ions must be considered. Few studies [33] have reported the application of Cu^2+^-containing coating on the AZ31 Mg alloy by use of the MAO method to minimize the corrosion rate of the alloy and assess its cytocompatibility. In the current work, the corrosion resistance and cytocompatibility of a Cu^2+^-coated AZ31 Mg alloy in a simulated body fluid (SBF) was evaluated to prepare a basis for the clinical usage of the biomedical AZ31 Mg alloy.

## 2. Materials and Methods

### 2.1. Specimen and MAO Preparation

An AZ31 Mg alloy sheet was mechanically cut into 25 mm × 20 mm × 8 mm specimens Afterward, the samples were sanded and polished using 1500 SiC abrasive papers to achieve a uniform roughness. They were cleaned ultrasonically first with acetone and then with ethanol for 10 min to degrease and then subjected to warm blow-drying. The MAO machine (WHYH-20) had a 20 kW DC power supply. A metal plate was used as a cathode, while AZ31 Mg alloy substrates were used as anodes. The experimental study was divided into three groups: AZ31 Mg alloy specified as the blank group (without coating), MgO-containing MAO coating sample specified as the control group, and Cu^2+^-containing coating sample specified as the experimental group. The MgO-containing MAO coating sample treated with the MAO process was prepared with 1.2 g/L of EDTA and 30 g/L of sodium silicate (Na_2_SiO_3_) dissolved in 1 L of deionized water by stirring. The Cu^2+^-containing coating sample treated with the MAO process was prepared with 1.0 g/L of cupric acetate Cu(CH_3_COO)_2_, 1.2 g/L of EDTA, and 30 g/L of sodium silicate (Na_2_SiO_3_) dissolved in 1 L of deionized water through stirring. The experimental parameters used are shown in Table 1. A water circulation system was used to keep the electrolyte temperature below 25 °C. After MAO treatment, all coated substrates were washed with distilled water and then subjected to warm blow-drying.

### 2.2. Coating Characterization

The cross-sectional morphologies and coating surfaces were tested using a scanning electron microscope (NSM-5600LV, JEOL, Tokyo, Japan). The samples were set inside an epoxy resin to observe the interface between coating and substrate. The specimen was ground and polished with a SiC abrasive paper up to 1500 grit. Subsequently, the scanning electron microscope was used to examine the cross-sectional microstructure. The porosity of the coating surface was counted using Image J software version 1.52 (National Institutes of Health, Bethesda, USA). A digital eddy current thickness gauge (1100, ElektroPhysik) was used to measure the coating thickness. The composition of the surface element in the coatings was measured using an energy-dispersive spectrometer (Hitachi S-4800, Hitachi, Ltd., Tokyo, Japan). The chemical composition of the coating surface was analyzed using an X-ray photoelectron spectroscope (PHI-5702, Mg KR X-ray, 1253.6 eV). An X-ray diffractometer (D/Max-2400, Rigaku Co., Ltd., Tokyo, Japan) was used to examine the phase composition of the coatings with Cu-Kα radiation by using 2θ ranging test angles of 10–90° and a step size of 0.02°.

### 2.3. Corrosion Behavior Test

#### 2.3.1. Electrochemical Measurements

An electrochemical system (Autolab PGSTAT302N) was used to perform a potentiodynamic polarization test for examining the corrosion characteristics of the AZ31 Mg alloy, MgO-containing coating, and Cu^2+^-containing coating. Simulated body fluid (SBF) was prepared, as justified by Kokubo and Takadama [34]. The electrochemical test took place in a well-designed tank that was completed using the three-electrode cell system. The specimen, the Ag/AgCl electrode (saturated with KCl), and platinum mesh served as the working, reference, and counter electrodes, respectively. The samples with an exposed zone of 0.5 cm^2^ were soaked in SBF for approximately 30 min before the test.; in order to obtain the open-circuit potential (OCP) voltage. The scan range was determined by the OCP plus or minus 1 V at a rate of 0.1 mV/s. The current density was calculated by Tafel approximation. All the electrochemical measurements were repeated in triplicate to ensure reproducibility.

#### 2.3.2. Long-Term Period Immersion Test

For the H_2_ evolution test, corrosion resistance was evaluated using a particular device at 37 °C. This device had an inverted funnel-burette type system and utilized a thermostatic oil bath, as described in a previous report [35]. The test was performed for 7 days, and the SBF solution was changed every 2 days. The corrosion rate was measured via H_2_ evolution. The test was repeated three times.

### 2.4. Cell Viability Test and Cell Morphology Examination

The cytotoxicity, cell morphology, and cell viability of the samples were tested using mouse fibroblast L-929 cells (Shanghai qishi biotechnology co., LTD, Shanghai, China). They were grown in RPMI-1640 in 37 °C humidified conditions and at an atmosphere of 5% CO_2_ supplemented with 10% (*v/v*) fetal bovine serum and 1% streptomycin and penicillin (*v/v*). The extraction medium of the examined materials was prepared using RPMI-1640 with a surface area/extraction medium ratio of 1 cm^2^/mL. Cells were incubated (5 × 10^4^ cells/well) in 24-well plates for 24 h to allow attachment. The cell culture medium was replaced with the extraction medium. CCK8 (Cell Counting Kit-8) assay was used to measure cell viability. This assay was conducted at 1, 2, and 4 days after incubation. After incubation for 2–4 h, the content was placed in a 96-well plate. Optical density value was calculated at 450 nm by using a microplate reader (Elx 800, Bio-Tek, Winooski, VT, USA), and the CCK8 assay was repeated three times. An unpaired single-tailed Student’s t-test was conducted to analyze the data statistically. Significance was set at the *p*-value of <0.05. The cell culture photos were captured using an inverted microscope (IX2-Olympus, Tokyo, Japan).

## 3. Results and Discussion

### 3.1. Sample Characterization

Figure 1 shows the visual appearance of MAO/AZ31 (Figure 1a1) and Cu/MAO/AZ31 (Figure 1b1). The MAO/AZ31 coating exhibited a white color, whereas the Cu/MAO/AZ31 coating had a light red color.

In Figure 1, both (a2) and (b2) display the morphological features determined via scanning electron microscopy (SEM) in MAO/AZ31 and Cu/MAO/AZ31, respectively. The coatings revealed microcracks and various pores with varying depth and other defects.

In Figure 1, both (a2) and (b2) refer to the features of the protective coatings of Mg alloys produced via MAO. The coating found in the conventional silicate electrolyte comprised typical MAO coating defects, including pores and cracks. The excess thermal stress caused microcracks during the fast melt solidification when contacting the coating surface and the electrolytic solution together [36]. However, the pores were obtained when gas bubbles and molten oxide were then released out to the coating surface through the discharge channels. During arc-generated melting, the pores were assumed to exist as “footprints” of the arc discharge channels that reached the electrolyte–coating interface [19]. At a high voltage, the sparking electrical discharge accompanied by electrical oxidation would form the structure of the pores. The inner structure of the oxide coating was indicated by the size, number, and distribution of pores. This structure was shown by the on-surface round pores after discharge and longitudinal pores among discharge products. The size and number of pores depend on a number of factors, including bath composition, alloy composition, current characteristics applied, and application time [37]. Figure 1b2 shows a smooth surface and few pores (with an average size of 19.3 μm), encompassing 1.39% of the total area, as analyzed by Image J software. By contrast, the MgO-containing MAO coating had a vast number of pores (with an average size of 28.6 μm), encompassing 2.57% of the total area in Figure 1a2. These pores were useful for cell attachment and anchorage in the outer implant layer [22].

The cross-sectional morphology in Figure 2 depicts the microstructures of the coatings, which exhibited pores in the outer surface film and a barrier in the inner surface film. The MAO/AZ31 coating in Figure 2a had a thickness of 16.2 ± 3 µm. Figure 2b shows that the Cu/MAO/AZ31 coating had a thickness of 26 ± 5 µm, probably due to the addition of Cu^2+^ ions to the solution. The different thicknesses of the coatings, that is, the Cu/MAO/AZ31 coating being thicker than the MAO/AZ31 coating, were due to the differences in conductivity of the electrolyte. The pores in the Cu/MAO/AZ31 coating were more regular than those in the MAO/AZ31 coating.

The chemical composition on the surfaces of MAO/AZ31 and Cu/MAO/AZ31 coatings was examined via energy-dispersive spectrometry (EDS) point analysis. The elements of O, Na, Mg, Al, and Si were detected in the MAO/AZ31 coating (Figure 3a). For the Cu/MAO/AZ31 coating, elements such as O, Na, Mg, Al, Si, and a trace amount of Cu were present throughout the coating (Figure 3b). The distribution of Cu^2+^ throughout the entire coating with a relatively low concentration indicated that the Cu^2+^ was incorporated successfully into the MAO coating. The Cu concentration was approximately 1.4 wt%, with 0.5 at% in the Cu/MAO/AZ31 coating. The other elements were comparable with those of the MAO/AZ31 coating.

The crystalline structures of the MAO/AZ31, and Cu/MAO/AZ31 coated surface were characterized using X-ray diffraction (XRD) patterns. Figure 4 presents the XRD pattern on the surface of both the substrate and the coated samples. Mg peaks were the most observed and comprised the highest peaks in all three samples. MgO existed in MAO/AZ31 and Cu/MAO/AZ31 coatings because of the MAO process represented in the three clear peaks [38], indicating that MAO/AZ31 formed in the conventional silicate electrolyte consisted mostly of MgO. All the peaks of the compounds containing Cu were invisible in the XRD spectrum of the Cu/MAO/AZ31 coating. A small amount of Cu could be detected by EDS due to the low amount of Cu^2+^ in the electrolyte and the fact that the content of the compounds containing Cu was excessively low to be detectable via XRD. The Cu/MAO/AZ31 coating did not have any special peaks, indicating the low amount and the formation of amorphous CuO on the surface [39].

The X-ray photoelectron spectroscopy (XPS) survey scans of Cu/MAO/AZ31 are shown in Figure 5a, and Cu 2p high-resolution scans are presented in Figure 5b. XPS measurements were necessary to understand the chemical composition of the outer layer of the Cu^2+^-containing coating. The existence of O, Al, C, Mg, Si, Zn, Na, N, and Cu was confirmed from the scans shown in Figure 5a. High-resolution XPS scans of Cu 2p (Cu 2p_3/2_, Cu 2p_1/2_) are presented in Figure 5b. The spectrum of Cu 2p in the binding energy consisted of two peaks at 933.2–933.4 eV (Cu 2p_3/2_) and 953.0 eV (Cu 2p_1/2_) [22], indicating the presence of Cu (II) [40]. O 1s in the binding energy with the peak at 529.6 eV suggested the existence of Cu–O bond in the form of CuO [40,41,42], and Cu 2p_3/2_ was mostly responsible for CuO [43]. This result confirmed the successful synthesis of the Cu^2+^-containing coating.

### 3.2. Corrosion Analysis

#### 3.2.1. Electrochemical Test

Typical potentiodynamic curves and the resulting data of AZ31, MAO/AZ31, and Cu/MAO/AZ31 are depicted in Table 2 and Figure 6. The corrosion potential (E_corr_) of AZ31 was measured at -1.59 V, and it shifted towards the positive direction at -1.53 V for the MAO/AZ31 coating. For the Cu/MAO/AZ31 coating, E_corr_ shifted even more toward the positive direction, with the most positive E_corr_ (-1.35 V) compared with the other samples. The corrosion current densities (I_corr_) were 4.27 × 10^−9^ A/cm^2^ for the Cu/MAO/AZ31 coating, 1.27 × 10^−8^ A/cm^2^ for the MAO/AZ31 coating, and 4.30 × 10^−6^ A/cm^2^ for the AZ31 Mg alloy. In comparison with AZ31, Cu/MAO/AZ31 showed an increase in E_corr_ by approximately 239 mV and a decrease in the I_corr_ value by 30 times. This result suggested that the Cu/MAO/AZ31 coating enhanced the corrosion resistance of AZ31. The decrease in the corrosion rate of the Cu^2+^-containing coating spontaneously decreased the corrosion ion concentration of the AZ31 substrate surface. Accordingly, the corrosion resistance of the Cu/MAO/AZ31 coating had increased more than those of the MAO/AZ31 coating and AZ31 specimen. The enhanced corrosion resistance of the MAO/AZ31 coating might be due to the high stability of MgO that occurred during MAO. Therefore, the Cu/MAO/AZ31 coating exhibited superior corrosion resistance to the MAO/AZ31 coating and AZ31 due to the existence of Cu^2+^ ions in the electrolyte that enhanced the thickness and reduced the porosity of the coating.

#### 3.2.2. H_2_ Evolution Test

Calculating the H_2_ volume generation to measure the corrosion rate of Mg alloys in an SBF solution during immersion provides accurate results. While 1 mol atom of dissolved Mg is equal to 1 mol of H_2_ gas, the quantity of H_2_ evolution is unaffected by the products of the corrosion on the Mg surface [44]. Figure 7a shows the H_2_ evolution test analysis of the uncoated AZ31 alloy, MAO/AZ31, and Cu/MAO/AZ31 substrates immersed in SBF for a continuous period. The total volumes of H_2_ released from the AZ31 alloy, MAO/AZ31, and Cu/MAO/AZ31 after 7 days of immersion were 6.6, 3, and 1.6 mL, respectively. AZ31 displayed the highest corrosion rate, followed by MAO/AZ31, and Cu/MAO/AZ31 showed the lowest corrosion rate. Cu/MAO/AZ31 was well-protected by Cu^2+^ ions via MAO and not easily destroyed by the aggressive ions in the SBF solution. The rate of H_2_ released from AZ31 was 0.053 mL/cm^2^/day, that from MAO/AZ31 was 0.0241 mL/cm^2^/day, and that from Cu/MAO/AZ31 was 0.0128 mL/cm^2^/day, which is close to the tolerable level of the H_2_ release rate in the human body (0.01 mL/cm^2^/day) [45]. These rates of H_2_ evolution were calculated using H_2_ evolution rate (mL/cm^2^/day) [46]. The H_2_ evolution rate can be related to the corrosion rate. Figure 7b shows the corrosion rates measured based on the H_2_ evolution rate by using the formula of CR (mm/year) = K × ∆H_2_ (mL/cm^2^/day) that has been explained in a previous report [44]. K should be calculated by converting into
(1)$$\left(\frac{365\;\mathrm{day}}{\rho .1\;\mathrm{yr}}\times \frac{1\;g}{1000\;\mathrm{mg}}\times \frac{10\;\mathrm{mm}}{1\;\mathrm{cm}}\right)$$
where ρ is the standard density (Mg is 1.74 g/cm^3^), which was calculated to be 0.111, 0.050, and 0.026 mm/year in AZ31, MAO/AZ31, and Cu/MAO/AZ31, respectively. MAO and Cu^2+^-containing coatings were able to shield the AZ31 Mg alloy in SBF, with the Cu/MAO/AZ31 coating showing a superior barrier property and acting as a protective layer. This result is consistent with those of the electrochemical analysis.

### 3.3. Cytotoxicity Test

The cytotoxicity of metals must be checked in vitro before in vivo, and CCK8 is a good choice for this purpose. The cytotoxic effects of AZ31, MAO/AZ31, and Cu/MAO/AZ31 were analyzed by measuring cell viability, as shown in Figure 8. A score of > 80% in viability of L-929 cells seeded with the extracts of AZ31, MAO/AZ31, and Cu/MAO/AZ31 for 1–4 days verified the nontoxicity of all samples mentioned above. From these results, we concluded that all the samples were cytocompatible. Cells cultured in Cu/MAO/AZ31 demonstrated the highest cell viability was comparable with the negative control group (*p* < 0.001) when cultured for 1, 2, and 4 days. All groups showed a significant difference (*p* < 0.001) compared with one another when cultured for 1, 2, and 4 days, except for the negative and MAO/AZ31 groups on day 4, showing *p* < 0.01 The Cu/MAO/AZ31 coating was bioactive and could promote the growth of L-929 in vitro. The Cu^2+^-containing coating exhibited acceleration for L-929 cell proliferation and differentiation. Hadidi et al. [47] reported an improved cell activity with Cu^2+^ ions and an increased osteoblast. Hadidi et al. and Marie [47,48] reported thinning of bone tissue after Cu^2+^ was missing due to decreased osteogenesis. Materials released from implants to body tissues should have excellent biocompatibility and must be nontoxic. On the contrary, Brewer and Lee et al. [49,50] reported that a high concentration of Cu^2+^ caused biological toxicity. The toxic effects of Cu^2+^ released after localized corrosion and selective dissolution from active metals were also reported; hence, they must be carefully controlled. We concluded that Cu^2+^ ions should be in a low concentration that is compatible with the needs of the human body and does not lead to cytotoxicity. However, further research is required for clinical applications, such as studies on in vivo antibacterial activity and compatibility.

The morphology of L-929 cells grown in RPMI-1640 media and seeded with extracts derived from (a) AZ31, (b) MAO/AZ31, (c) Cu/MAO/AZ31, and (d) control group (negative) was observed under an inverted microscope (20×), as shown in Figure 9. The highest cell viability was noted in Cu/MAO/AZ31, whereas AZ31 had the lowest cell viability. The cell morphology in the extracts of Cu/MAO/AZ31 was as normal and healthy as in the negative control group. Cu/MAO/AZ31 presented good cytocompatibility, suggesting that the material was not toxic for cell growth.

## 4. Conclusions

A uniform Cu-free coating (MAO/AZ31) and a Cu-containing coating (Cu/MAO/AZ31) were coated on a biodegradable AZ31 Mg alloy. They were successfully achieved through MAO in a basic silicate electrolyte solution. The structure, composition, corrosion resistance, and cytotoxicity of the coatings were estimated. The following conclusions were obtained:The Cu-free electrolyte consisting of EDTA and Na_2_SiO_3_ resulted in a MgO-containing coating (MAO/AZ31). The Cu-containing electrolyte comprising EDTA, Na_2_SiO_3_, and Cu (CH_3_COO)_2_ led to a Cu^2+^-containing coating (Cu/MAO/AZ31). The porosity in MAO/AZ31 increased in quantity with wide pores, but the thickness decreased. By contrast, Cu/MAO/AZ31 showed minimal pores with an increased thickness.Electrochemical measurements and the H_2_ evolution test showed that the corrosion rate of Cu/MAO/AZ31 decreased compared with that of MAO/AZ31. Additionally, there will be a decrease in the rate of degradation on the AZ31 Mg alloy.In vitro cytotoxicity examination of Cu/MAO/AZ31 confirmed that a cytotoxic reaction to L-929 cells was not induced; however, cell proliferation was established during the first 4 days. Further research is needed for clinical applications.These observations indicated that the Cu^2+^-coated AZ31 Mg alloy has excellent corrosion resistance and cytocompatibility, giving it the potential to be used as a medical implant material such as miniplates for the purpose of skeletal anchorage.

## Figures and Tables

**Figure 1 materials-13-02647-f001:**
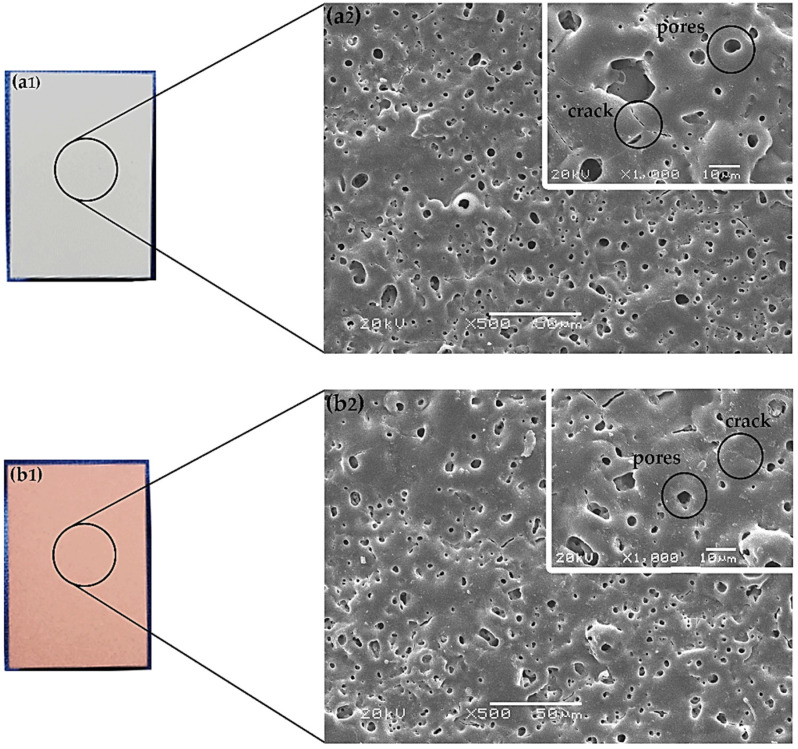
Optical morphology of (**a1**) MAO/AZ31, and (**b1**) Cu/MAO/AZ31; SEM morphology of (**a2**) MAO/AZ31, and (**b2**) Cu/MAO/AZ31.

**Figure 2 materials-13-02647-f002:**
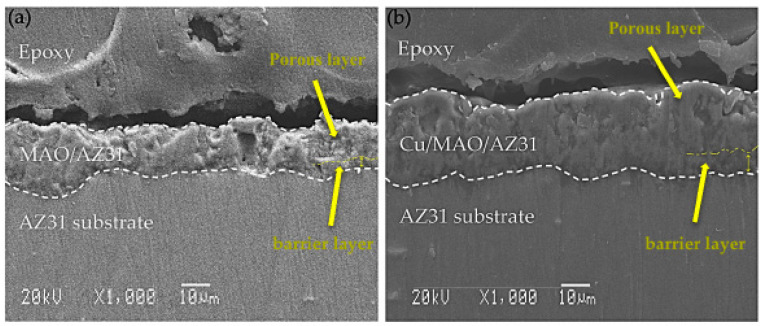
SEM cross-sectional images of (**a**) MAO/AZ31, (**b**) Cu/MAO/AZ31.

**Figure 3 materials-13-02647-f003:**
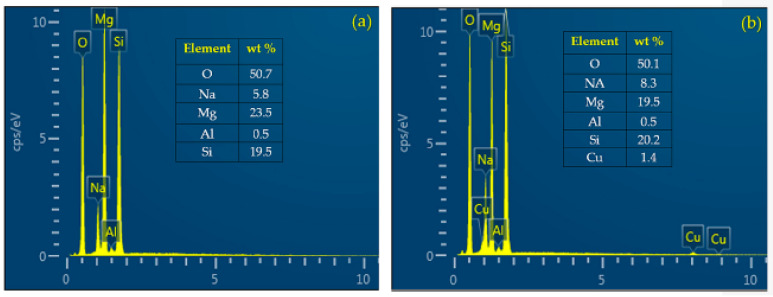
EDS point analysis of (**a**) MAO/AZ31 and (**b**) Cu/MAO/AZ31.

**Figure 4 materials-13-02647-f004:**
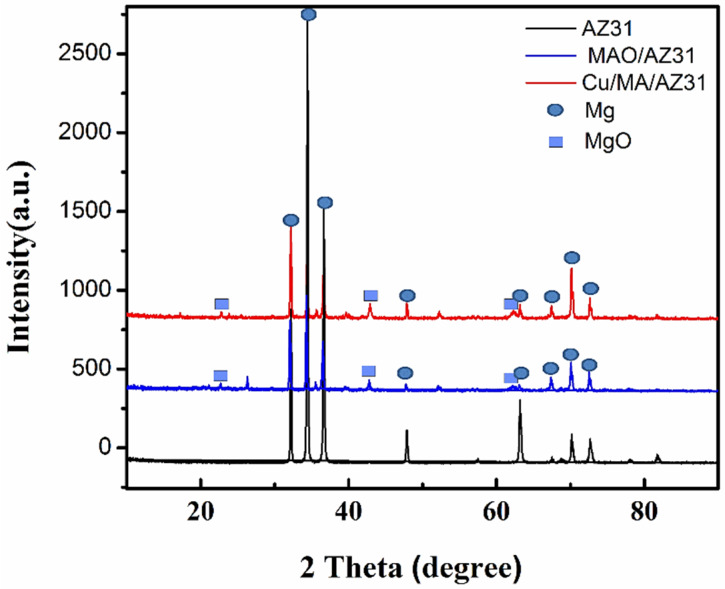
XRD patterns of AZ31, MAO/AZ31, and Cu/MAO/AZ31.

**Figure 5 materials-13-02647-f005:**
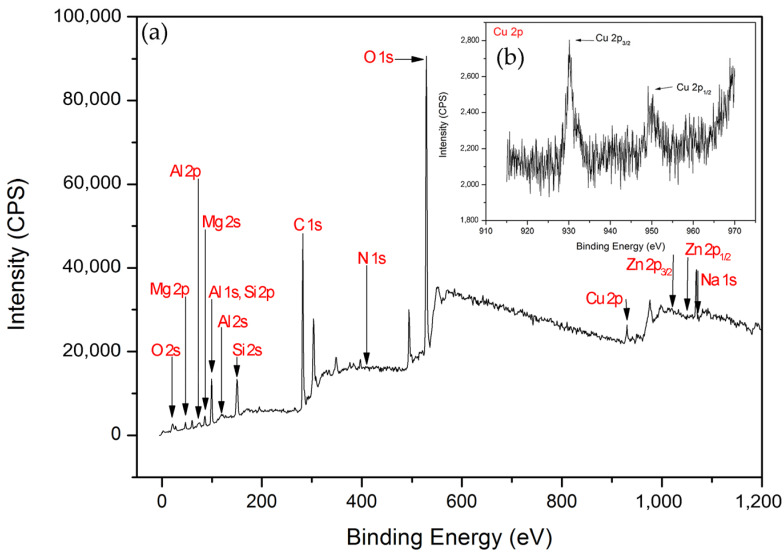
(**a**) XPS survey scans of Cu/MAO/AZ31 coating, (**b**) XPS spectra of Cu 2p.

**Figure 6 materials-13-02647-f006:**
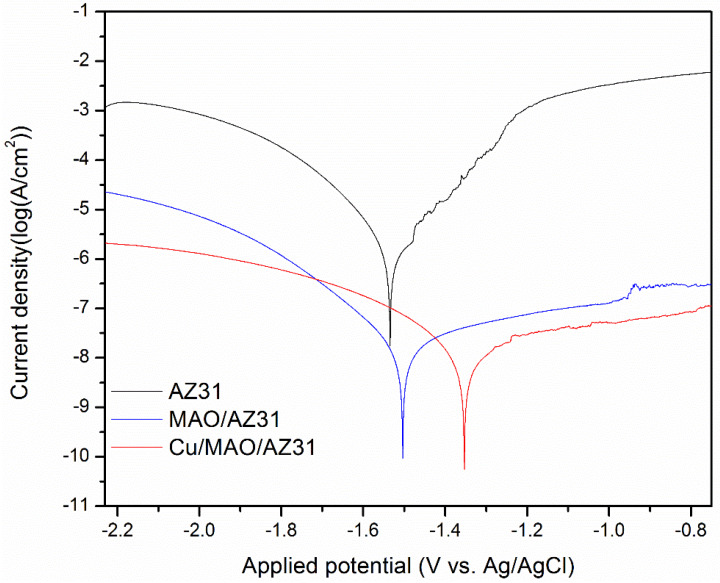
Typical potentiodynamic curves and resulting data of AZ31, MAO/AZ31, and Cu/MAO/AZ31 in a simulated body fluid (SBF) solution.

**Figure 7 materials-13-02647-f007:**
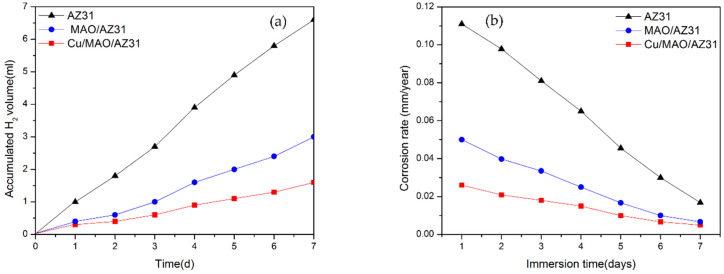
(**a**) H_2_ evolution test for substrates immersed in SBF for 7 days; (**b**) corresponding corrosion rate.

**Figure 8 materials-13-02647-f008:**
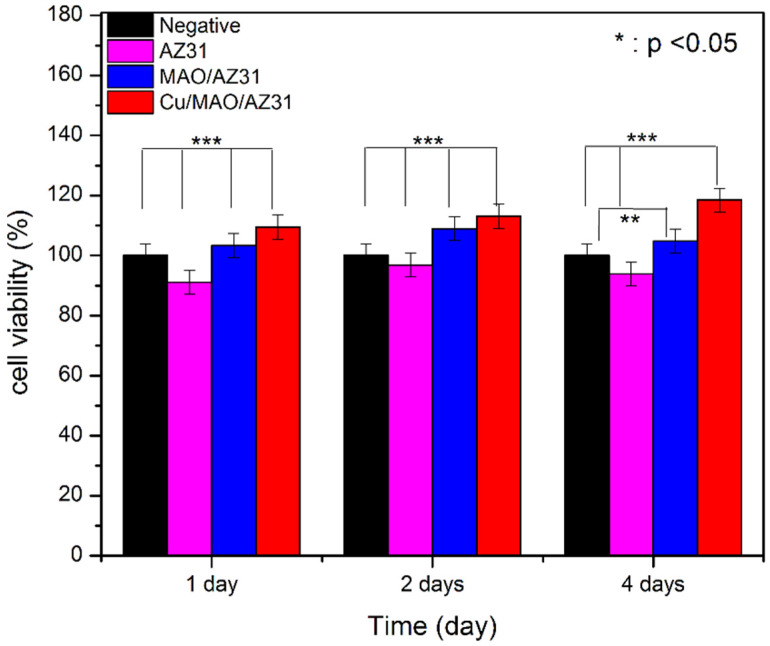
Cell viability of AZ31, MAO/AZ31, and Cu/MAO/AZ31 specimens.

**Figure 9 materials-13-02647-f009:**
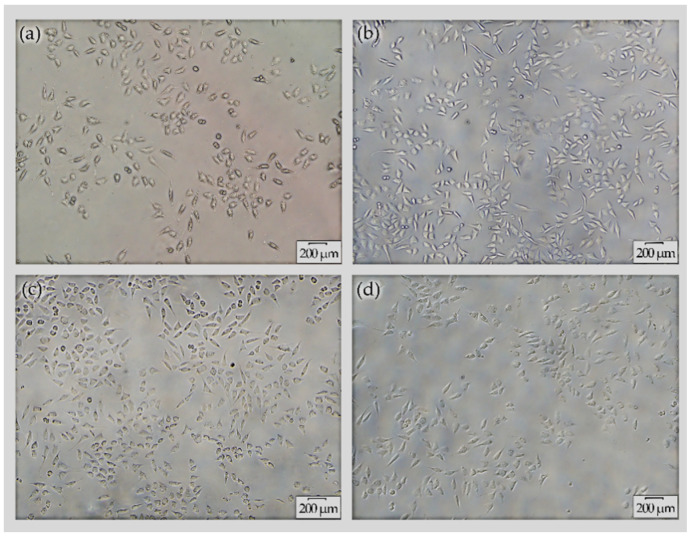
Cell morphology under inverted microscope (20×) for L-929 cells cultured with extracts from (**a**) AZ31, (**b**) MAO/AZ31, (**c**) Cu/MAO/AZ31, and (**d**) the negative group for 4 days.

**Table 1 materials-13-02647-t001:** Experimental parameters and electrolyte concentration during microarc oxidation (MAO).

Group	Electrolyte Concentration (g·L^−1^)	Current Frequency (Hz)	Positive Voltage (V)	Negative Voltage (V)	Current Density (A/dm^2^)	Process Time (min)
Control group	EDTA = 1.2, Na_2_SiO_3_ = 30.	150	410	75	1:1	10
Experimental group	Cu(CH_3_COO)_2_ = 1, EDTA = 1.2, Na_2_SiO_3_ = 30.	150	400	70	1:1	10

**Table 2 materials-13-02647-t002:** Parameters for the Tafel polarization method.

Samples	E_corr_ (V)	I_corr_ (A/cm^2^)
AZ31	−1.59	4.30 × 10^−6^
MAO/AZ31	−1.53	1.27 × 10^−8^
Cu/MAO/AZ31	−1.35	4.27 × 10^−9^

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
