# Peer review of "Influence of Cu2+ Ions on the Corrosion Resistance of AZ31 Magnesium Alloy with Microarc Oxidation"

_materials, 2020, doi:10.3390/ma13112647_

Round 1

Reviewer 1 Report

  1. Sentence: “….AZ31 Mg alloy, and to obtain a standard for the biomedical use of AZ31 Mg alloy in clinical applications” – What clinical applications?? If the author means resorbable implants - that's the wrong way. AZ31 main alloying additive is ALUMINUM which is toxic to human body. It is necessary to specify the possible range of applications for this alloy.

  1. Sentence: “The solution of the immersion test was renewed every 2 days for about 3 months in a beaker at 37 ËšC.” – Why? why every two days

  1. Figure 1. The visual appearance – I do not understand this presentation with colors. What's this all about? what is the purpose of this presentation ??

  1. Fig. 7 - very blurred. In this form they bring nothing to work.

5.Fig.8 - poor quality - please improve. And one question. How was the corrosion rate calculated in mm / year based on the amount of hydrogen released? Please explain. – THIS SENTENCE: “Figure (8b) shows the corrosion rates measured from the volume of the released H2 by the formula of CR (mm/yr) =K×ΔH2(mL/cm2/day) that was explained in the previous report [45]. – does not explain anything. Please do not refer to previous studies but explain step by step in this work.

  1. Figure 9. – poor quality – please improve

  1. Figure 10. – poor quality – please improve , too bright photos

  1. Table 2. Tafel polarization method parameters. - There are only two parameters in the table WHICH ARE NOT ANY KIND OF TAFEL POLARIZATION.

  Why was the resistance of polarization not calculated? corrosion current density? These parameters are very helpful for the analysis of corrosion behavior of magnesium alloys. Please complete this.

  1. This conclusion: "These observations indicate that Cu2 + -containing coated AZ31 magnesium alloy has excellent corrosion resistance and cytocompatibility, with a potential use as a medical implant material." - what uses examples concretely ??

Author Response

Response to Reviewer 1 Comments

Responses to Reviewer 1:

Thank you very much for your advice. We are very sorry for our carelessness. All the writing mistakes have been modified in the manuscript. The following responses are the answers for the specific problems.

Point 1: Sentence: “….AZ31 Mg alloy, and to obtain a standard for the biomedical use of AZ31 Mg alloy in clinical applications” – What clinical applications?? If the author means resorbable implants - that's the wrong way. AZ31 main alloying additive is ALUMINUM which is toxic to human body. It is necessary to specify the possible range of applications for this alloy.

Response 1: Thanks for your careful checks. According to the Reviewer’s comment and a further consideration, we thought that the original purpose was not reasonable according to the present characterizations. The purpose of this work should be limited to reduce the corrosion rate of AZ31 Mg alloy and prepare the biocompatible MAO coating and. According to that we have made correction.

The clinical usage was also explained in the manuscript in line 36-41.

 “Implants are used to reconstruct the failed tissue and bone fixation surgery, therefore a traditional biomaterial frequently needs a second surgery to remove it [1]. The skeletal-anchorage in orthodontics such as miniscrews and miniplates made of magnesium alloys provided stable implant materials that degraded in vivo [2, 3], eliminating the need for a second operation for implant removal will help to overcome the limitations of conventional orthodontic techniques [4].”

[1] Zhang, L., et al., Advances in microarc oxidation coated AZ31 Mg alloys for biomedical applications. Corrosion Science, 2015. 91: p. 7-28.

[2] Witte, F., et al., In vivo corrosion of four magnesium alloys and the associated bone response. Biomaterials, 2005. 26(17): p. 3557-3563.

[3] Chaya, A., et al., In vivo study of magnesium plate and screw degradation and bone fracture healing. Acta biomaterialia, 2015. 18: p. 262-269.

[4] Al-Dumaini, A.A., et al., A novel approach for treatment of skeletal Class II malocclusion: Miniplates-based skeletal anchorage. American Journal of Orthodontics and Dentofacial Orthopedics, 2018. 153(2): p. 239-247.

Point 2: Sentence: “The solution of the immersion test was renewed every 2 days for about 3 months in a beaker at 37 ËšC.” – Why? why every two days

Response 2: According to the Reviewer’s comment, the SBF solution was renew every 2 days to maintain pH=7.4 and ion concentration. According to this reference it was made [5].

[5] Fu, L., et al., Preparation and Characterization of Fluoride-Incorporated Plasma Electrolytic Oxidation Coatings on the AZ31 Magnesium Alloy. Coatings, 2019. 9(12): p. 826.

Point 3: Figure 1. The visual appearance – I do not understand this presentation with colors. What's this all about? what is the purpose of this presentation??

Response 3: According to the Reviewer’s comment, figure 1 shows the optical morphology of each coating. The colors are the natural characteristics of the two type coatings. The purpose of this presentation is to show the difference of the visual appearance between them.

Point 4: Fig. 7 - very blurred. In this form they bring nothing to work.

Response 4: Thanks for your careful checks. We are sorry for our carelessness. But according to the other Reviewer’s comment, this part has been deleted.

Point 5: 5. Fig.8 - poor quality - please improve. And one question. How was the corrosion rate calculated in mm / year based on the amount of hydrogen released? Please explain. – THIS SENTENCE: “Figure (8b) shows the corrosion rates measured from the volume of the released H2 by the formula of CR (mm/yr) =K×ΔH2(mL/cm2/day) that was explained in the previous report [45]. – does not explain anything. Please do not refer to previous studies but explain step by step in this work.

Response 5: Thanks for your careful checks. We are sorry for our carelessness. Based on your comments, we have made corrections. The figure has been improved and the sentence (line 245-251) has been rearranged to “The rate of H2 released from AZ31 was 0.053 mL/cm2/day, that from MAO/AZ31 was 0.0241 mL/cm2/day, and that from Cu/MAO/AZ31 was 0.0128 mL/cm2/day, which is close to the tolerable level of the H2 release rate in the human body (0.01 mL/cm2/day) [6]. These rates of H2 evolution were calculated using H2 evolution rate (mL/cm2/day)[7]. The H2 evolution rate can be related to the corrosion rate. Figure (7b) shows the corrosion rates measured based on the H2 evolution rate by using the formula of CR (mm/yr) = K × âˆ†H2 (mL/cm2/day) that has been explained in a previous report [8]”.

[6] Song, G., Control of biodegradation of biocompatable magnesium alloys. Corrosion science, 2007. 49(4): p. 1696-1701.

[7] Zhao, M.-C., et al., Influence of the β-phase morphology on the corrosion of the Mg alloy AZ91. Corrosion Science, 2008. 50(7): p. 1939-1953.

[8] Jang, Y., et al., Systematic understanding of corrosion behavior of plasma electrolytic oxidation treated AZ31 magnesium alloy using a mouse model of subcutaneous implant. Materials Science and Engineering: C, 2014. 45: p. 45-55.

Point 6: Figure 9. – poor quality – please improve

Response 6: Thanks for your careful checks. We are sorry for our carelessness. Based on your comments, we have made corrections.

Point 7: Figure 10. – poor quality – please improve, too bright photos

Response 7: Thanks for your careful checks. We are sorry for our carelessness. Based on your comments, we have made corrections.

Point 8: Table 2. Tafel polarization method parameters. - There are only two parameters in the table WHICH ARE NOT ANY KIND OF TAFEL POLARIZATION.

  Why was the resistance of polarization not calculated? corrosion current density? These parameters are very helpful for the analysis of corrosion behavior of magnesium alloys. Please complete this.

Response 8: Thanks for your kind comment. It is common to judge the corrosion performance of MAO coatings by the parameters of corrosion potential (Ecorr) and corrosion current density(icorr). They are considered as indicators in thermodynamics and in dynamics respectively. So, we circumspectly believe that Ecorr and icorr are sufficient.

Point 9: This conclusion: "These observations indicate that Cu2 + -containing coated AZ31 magnesium alloy has excellent corrosion resistance and cytocompatibility, with a potential use as a medical implant material." - what uses examples concretely??

Response 9: We are very sorry for our unclear expression, according to the Reviewer’s comment, the uses examples were given concretely, and we have made corrections to “These observations indicated that the Cu2+- coated AZ31 Mg alloy had excellent corrosion resistance and cytocompatibility, with a potential use as a medical implant material such as miniplates for the skeletal anchorage purpose”.

Special thanks to you for your good comments!

Reviewer 2 Report

General comments

English grammar and syntax should be strongly improved.

An huge amount of typesetting errors are present throughout the text.

Introduction Section

Page 2, lines 40-46. Information about biodegradable implants does not fit the topic of this work, since all materials investigated cannot be considered biodegradable. I suggest to delete this part of the introduction.

Materials and Methods

Page 3, line 98 and Table 1. The reported chemical formula of cupric acetate is wrong.

Page 3, Table 2. Reported values of current density are unclear.

Page 4, lines 108-109: it is not clear what the "interference between the coating and the substrate" is.

Page 4 line 111: Information about the ImageJ software are missing (vendor? version?)

Page 4 line 114: Chemical composition of what?

Page 4 line 123: the meaning of the acronym SBF should be reported.

Page 5 line 147: it is not clear what CCK8 is used for.

Results and Discussion section

Page 5, Figure 1: It is not clear whether Figures 1a and 1b are pictures or drawings. Anyway, they do not show any useful information. I suggest to delete them.

Page 6, lines 177-180. How have the results been obtained?

Page 6 line 182: it is not clear where the "pores in the outer surface film" and the "barrier in inner surface film" are. I suggest to indicate them in Figure 2.

Page 6 lines 187-191. This part is very confusing. The authors are discussing about the effect of coatings features on corrosion resistance without showing any evidence or results to support their conclusions.

Page 7 lines 198-199. The sentence "The distribution of … is completely oxidized" sounds completely unclear to me.

Page 7 lines 205-215. The XRD analysis shown in Figure 4 is not convincing to me. The authors state that samples contain both elemental Mg and MgO. That sounds very weird to me. In addition, Figure 3 shows that coatings contain a significant amount of Si. Where is it in Figure 4? Lastly, there is no evidence that Cu has an amorphous nature.

Figure 5: For the sake of comparison, XPS should be reported also for MAO/AZ31 coating.

Page 11. Lines 283-284. It is not clear how was the rate of H2 evolution calculated.

Author Response

Response to Reviewer 2 Comments

Responses to Reviewer 2:

Thank you very much for your advice. We are very sorry for our carelessness. All the writing mistakes have been modified in the manuscript. The following responses are the answers for the specific problems.

General comments

Point 1: English grammar and syntax should be strongly improved.

Response 1: Thanks for your careful checks. We feel sorry for our poor writings. However, we invited a native English speaker to polish our article. Due to his help, the article had been edited extensively. And we hope the revised manuscript could be acceptable.

Point 2: A huge amount of typesetting errors are present throughout the text.

Response 2: Thanks for your careful checks. We feel sorry for our poor writings. Based on your comments, we have made corrections.

Introduction Section

Point 1: Page 2, lines 40-46. Information about biodegradable implants does not fit the topic of this work, since all materials investigated cannot be considered biodegradable. I suggest to delete this part of the introduction.

Response 1: Thanks for your suggestion, based on your comments, we have deleted the sentence.

Even though this part was deleted, in this work, the coatings’ corrosion behaviors were characterized by hydrogen evolution examination in SBF, besides Tafel test. Simultaneously, the cell test also confirmed the coatings’ biocompatibility. So, we circumspect believe that the biodegradability fits this topic somewhere in the manuscript. Also, more explanation was written about the clinical usage in line 36-41.

Materials and Methods

Point 1: Page 3, line 98 and Table 1. The reported chemical formula of cupric acetate is wrong.

Response 1: Thanks for your careful checks. We are sorry for our carelessness. Based on your comments, we have made corrections.

Point 2: Page 3, Table 2. Reported values of current density are unclear.

Response 2: We are sorry for our unclear expression. Table 2 lists the corrosion potentials and corrosion current densities calculated results of Tafel tests, which was mentioned in Page 9, line (219-221).

Point 3: Page 4, lines 108-109: it is not clear what the "interference between the coating and the substrate" is.

Response 3: Thanks for your careful checks. We are sorry for our carelessness. Based on your comments, we have made corrections “interface between the coating and the substrate”.

Point 4: Page 4 line 111: Information about the ImageJ software are missing (vendor? version?)

Response 4: Thanks for your careful checks. We are sorry for our carelessness. Based on your comments, The ImageJ software was open-access. The version was added in the manuscript as Image J (Ver. 1.52).

Point 5: Page 4 line 114: Chemical composition of what?

Response 5: We are very sorry for our unclear expression and we have made corrections to “the chemical composition of the coating surface”.

Point 6: Page 4 line 123: the meaning of the acronym SBF should be reported.

Response 6: Thanks for your help, we feel really sorry for our carelessness. Based on your comments, we have made corrections.

Point 7: Page 5 line 147: it is not clear what CCK8 is used for.

Response 7: We are very sorry for our unclear expression and we have made corrections to “The CCK8 (Cell Counting Kit-8) assay was used to measure cell viability”.

Results and Discussion section

Point 1: Page 5, Figure 1: It is not clear whether Figures 1a and 1b are pictures or drawings. Anyway, they do not show any useful information. I suggest to delete them.

Response 1: According to the Reviewer’s comment, Figures 1a and 1b are pictures. figure 1a and 1b shows the optical morphologies of each coating. The colors are the natural characteristics of the two type coatings. The purpose of this presentation is to show the difference of the visual appearance between them.

Point 2: Page 6, lines 177-180. How have the results been obtained?

Response 2: According to the Reviewer’s comment, the pores have been calculated using Image J software (Ver. 1.52).

Point 3: Page 6 line 182: it is not clear where the "pores in the outer surface film" and the "barrier in inner surface film" are. I suggest to indicate them in Figure 2.

Response 3: Thanks for your suggestion, according to the Reviewer’s comment, the outer and inner surface film have been indicated in Figure 2.

Point 4: Page 6 lines 187-191. This part is very confusing. The authors are discussing about the effect of coatings features on corrosion resistance without showing any evidence or results to support their conclusions.

Response 4: Thanks for your help, we feel really sorry for our carelessness and we have deleted the sentence.

Point 5: Page 7 lines 198-199. The sentence "The distribution of … is completely oxidized" sounds completely unclear to me.

Response 5: We are very sorry for our unclear expression and we have made corrections to “The Cu2+ in the Cu2+- containing coating has been distributed throughout the whole coating with relatively low concentration, indicating that the Cu2+ was incorporated successfully into the MAO coating”.

Point 6: Page 7 lines 205-215. The XRD analysis shown in Figure 4 is not convincing to me. The authors state that samples contain both elemental Mg and MgO. That sounds very weird to me. In addition, Figure 3 shows that coatings contain a significant amount of Si. Where is it in Figure 4? Lastly, there is no evidence that Cu has an amorphous nature.

Response 6: We are very sorry for our unclear expression and we have made corrections. FOR XRD, we have reorganized the words. The signal of elemental Mg was from the substrate.

For Si, in relevant references [1], Si was detected by EDS in a quite high level but undetected by XRD. They are also attribute this to the amorphous Si-compounds.

The Cu/MAO/AZ31 coating did not have any special peaks, indicating the low amount and the formation of amorphous CuO on the surface [2] . Additionally, in XPS, Cu element exist in the form of CuO.

[1] Krząkała, A., A. Kazek-Kęsik, and W. Simka, Application of plasma electrolytic oxidation to bioactive surface formation on titanium and its alloys. RSC advances, 2013. 3(43): p. 19725-19743.

[2] Patake, V., et al., Electrodeposited porous and amorphous copper oxide film for application in supercapacitor. Materials Chemistry and Physics, 2009. 114(1): p. 6-9.

Point 7: Figure 5: For the sake of comparison, XPS should be reported also for MAO/AZ31 coating.

Response 7: Thanks for your suggestion. According to the Reviewer’s comment, we have done XPS to prove the presence of Cu (II) only.

Point 8: Page 11. Lines 283-284. It is not clear how was the rate of H2 evolution calculated.

Response 8: According to the reviewer’s comments, one mole of hydrogen gas is equal to one mole atom of magnesium dissolved. So, the hydrogen evolution rate, VH [mL/(cm2.day)] was measured by calculating the total amount of H2 released dividing the area of the sampled multiplying the days. Line (248-249).

Special thanks to you for your good comments!

Reviewer 3 Report

Comments to the Authors:
The authors of this paper present an evaluation of the corrosion resistance of Cu^{2+}-containing coated AZ31 Mg alloy in a simulated body fluid for the preparation of a basis of biomedical AZ31 Mg alloy for clinical usage. It is an interesting work, nevertheless, some details should be considered by the authors:

Introduction

COMMENT: Page 1, line 36: I suggest the authors to add more recent references.

COMMENT: Page 2, line 46: I suggest the authors to add more recent references.

Results and discussion

COMMENT: Page 5, Fig. 1: I suggest the authors to improve the quality of the images shown in Fig. 1 (the scale and the details are hardly seen).

COMMENT: Page 6, lines 185-186 (“Providing different thickness… conductivity of the voltage”): This conductivity difference hypothesis may be further commented/explained.

COMMENT: Page 7, line 213: What is the detection limit of the XRD devise used?

The reported data are sufficiently discussed and commented and the results support sufficiently the authors conclusions. Therefore, I think that this paper is suitable for publication.

Author Response

Response to Reviewer 3 Comments

Responses to Reviewer 3:

Thank you very much for your advice. We are very sorry for our carelessness. All the writing mistakes have been modified in the manuscript. The following responses are the answers for the specific problems.

Introduction

Point 1: Page 1, line 36: I suggest the authors to add more recent references.

Response 1: Thanks for your suggestion, according to the reviewer’s comments, some recent references have been added.

Point 2: Page 2, line 46: I suggest the authors to add more recent references.

Response 2: Thanks for your suggestion, but according to other reviewer’s comments, this part has been deleted.

Results and discussion

Point 1: Page 5, Fig. 1: I suggest the authors to improve the quality of the images shown in Fig. 1 (the scale and the details are hardly seen).

Response 1: Thanks for your suggestion. Based on your comments, we have made corrections.

Point 2: Page 6, lines 185-186 (“Providing different thickness… conductivity of the voltage”): This conductivity difference hypothesis may be further commented/explained.

Response 2: Thanks for your careful checks. We are sorry for our carelessness. Based on your comments, the sentence has changed to “the conductivity of the electrolyte”.

If the conductivity increased then the voltage will decrease at the same current density.

The coating thickness increased with increasing voltages, this occurred because the increase in voltage led to the increase of the potential of oxidation and consequently the increment of the MAO coating thickness [1].

[1] Zhang, L., et al., Advances in microarc oxidation coated AZ31 Mg alloys for biomedical applications. Corrosion Science, 2015. 91: p. 7-28.

Point 3: Page 7, line 213: What is the detection limit of the XRD devise used?

Response 3: We are very sorry for our ignorance of the detection limit in XRD. However, the 1.4 wt% Cu is equal to about 0.4 at%. Generally, XRD cannot respond to such a trace phase.

Special thanks to you for your good comments!

Reviewer 4 Report

The paper deals with surface treatment of AZ31 using micro-arc oxidation in an electrolyte containing Cu(II) ions. Obtained MgO-based coatings showed better corrosion resistance than reference coatings produced in an electrolyte without Cu(II). The study is well designed and documented and results are correctly discussed. I recommend the paper for publication in Materials. Still, the manuscript should be improved somewhat before publication; see my comments below.

  • I recommend the authors using a service of a professional proof-reader. The English needs to be improved. Some sentences are not easy to understand and confusing.
  • The chapter on electrochemical measurements needs to be improved. It is not explained in the experimental part how the potentiodynamic curves were measured. What potential it started on? What was the scan rate? How the corrosion current at OCP was determined?
  • The part dealing with the immersion test is little informative. In photographs in Figure 7, there is nothing to see. The evaluation is very qualitative. A quantification needs to be added, e.g., mass changes, colour changes, number of pits, etc. If not possible, this section should be fully removed. The hydrogen release measurements are good enough.
  • There must be a mistake in Figure 8. Based on Figure 8(a), the corrosion rate was about constant in time. How is it possible that it is decreasing strongly in (b)? Please, correct.
  • The conclusions should be made a bit more straightforward and less wordy. E.g., in line 341-342, the statement “while the corrosion resistance of Cu/MAO/AZ31 was increased” is redundant. The same is said in the previous sentence.

Further minor remarks follow.

  • Line 47 and further. Elements and chemical formula should not begin with capital letters.
  • Line 75 and elsewhere. It is not usual to refer to papers including authors’ first names. E.g., Zaki et al. is sufficient.
  • Line 76. Experimental Cu coatings, I suppose.
  • Line 75-78. The first sentence states the Cu(II) addition improved the corrosion resistance. Why do you start the second sentence, which says the same, by “on the other hand”?
  • Line 79. You state that “few studies have reported the application…”. These have to be referred to.
  • Line 92. An electrolyte is an electrolyte, not a cathode. As a cathode, a metal plate or similar is usually used. Please, change it.
  • Line 98 and Table 1. Cupper(II) acetate’s formulae is given incorrectly. It should be Cu(CH3COO)2 or Cu(CO2CH3)2.
  • Table 1. Can you comment on the fact that the voltage was not identical for Cu-containing and Cu-free baths? Could it affect the results? Please, discuss this in the paper.
  • Line 109. It was not the epoxy resin, but the specimen to be polished.
  • Line 127. It is not explained how the samples were prepared for the measurement. How were they electrically connected? How was the working area delimitated?
  • Line 128: The OCP should settle, or was it really done? Please, change the expression.
  • Line 135-136. Was the device conducted, or the experiment?
  • Line 169-170. Were the pores released, or was it hydrogen which was released, forming the pores?
  • Line 184. Keep the same number of valid digits, i.e., 26 +- 5.
  • Line 186. Voltage has no conductivity. Change the sentence.
  • Line 187-191. You are discussing results that were not shown yet. You should not talk about corrosion resistance in this section. It should be done after corrosion data are presented.
  • Line 200, 219. If formulated as it is, the word respectively does not belong to the sentence.
  • Line 206-207. Please, improve the sentence. It is hard to understand.
  • Line 223. Carbon can come from Cu(II) acetate as well.
  • Line 234 and elsewhere. You are referring to the open circuit potential or free corrosion potential. Corrosion potential is any potential a corroding system can reach. The same applies for the corrosion current.
  • Line 235. It makes no sense to give the Eocp with so high preciseness. You cannot get so precise value reproducibly.
  • Line 240. It is impossible to define the increase this precisely. It seems funny in combination with the word “about”.
  • Line 282-283. The number of digits is too high regarding the sensitivity of the measurement. Adjust it in view of the standard deviation, which should anyway be given.

Author Response

Response to Reviewer 4 Comments

Responses to Reviewer 4:

Thank you very much for your advice. We are very sorry for our carelessness. All the writing mistakes have been modified in the manuscript. The following responses are the answers for the specific problems.

Point 1: I recommend the authors using a service of a professional proof-reader. The English needs to be improved. Some sentences are not easy to understand and confusing.

Response 1: Thanks for your careful checks. We feel sorry for our poor writings. However, we invited a native English speaker to polish our article. Due to his help, the article had been edited extensively. And we hope the revised manuscript could be acceptable.

Point 2: The chapter on electrochemical measurements needs to be improved. It is not explained in the experimental part how the potentiodynamic curves were measured. What potential it started on? What was the scan rate? How the corrosion current at OCP was determined?

Response 2: Thanks for your help, according to the Reviewer’s comment, the OCP was measured after the sample emerged in SBF for 0.5h. The scan range was determined by the OCP plus or minus 1 V in a rate of 0.1 mV/s. The current density was calculated by Tafel approximation.

Point 3: The part dealing with the immersion test is little informative. In photographs in Figure 7, there is nothing to see. The evaluation is very qualitative. A quantification needs to be added, e.g., mass changes, colour changes, number of pits, etc. If not possible, this section should be fully removed. The hydrogen release measurements are good enough.

Response 3: According to the Reviewer’s comments, this part has been deleted.

Point 4: There must be a mistake in Figure 8. Based on Figure 8(a), the corrosion rate was about constant in time. How is it possible that it is decreasing strongly in (b)? Please, correct.

Response 4: Thanks for your help, according to the reviewer’s comments, we think that the figure matches the calculation and nothing wrong with it. The two figures have a good correlation.

Point 5: The conclusions should be made a bit more straightforward and less wordy. E.g., in line 341-342, the statement “while the corrosion resistance of Cu/MAO/AZ31 was increased” is redundant. The same is said in the previous sentence.

Response 5: Thanks for your advice, based on your comments, we have made corrections.

Further minor remarks follow.

Point 6: Line 47 and further. Elements and chemical formula should not begin with capital letters.

Response 6: Thanks for your careful checks. We are sorry for our carelessness. Based on your comments, we have made corrections.

Point 7: Line 75 and elsewhere. It is not usual to refer to papers including authors’ first names. E.g., Zaki et al. is sufficient.

Response 7: Thanks for your careful checks. We are sorry for our carelessness. Based on your comments, we have made corrections.

Point 8: Line 76. Experimental Cu coatings, I suppose.

Response 8: Thanks for your careful checks. We are sorry for our carelessness. Based on your comments, we have made corrections.

Point 9: Line 75-78. The first sentence states the Cu(II) addition improved the corrosion resistance. Why do you start the second sentence, which says the same, by “on the other hand”?

Response 9: Thanks for your careful checks. We are sorry for our carelessness. Based on your comments, we have made corrections.

Point 10: Line 79. You state that “few studies have reported the application…”. These have to be referred to.

Response 10: Thanks for your advice, Based on your comments, we have added a reference [1].

[1] Chen, J., et al., In vitro degradation and antibacterial property of a copper-containing micro-arc oxidation coating on Mg-2Zn-1Gd-0.5 Zr alloy. Colloids and Surfaces B: Biointerfaces, 2019. 179: p. 77-86.

Point 11: Line 92. An electrolyte is an electrolyte, not a cathode. As a cathode, a metal plate or similar is usually used. Please, change it.

Response 11: Thanks for your careful checks. We are sorry for our carelessness. Based on your comments, we have made corrections.

Point 12: Line 98 and Table 1. Cupper(II) acetate’s formulae is given incorrectly. It should be Cu(CH3COO)2 or Cu(CO2CH3)2.

Response 12: Thanks for your careful checks. We are sorry for our carelessness. Based on your comments, we have made corrections.

Point 13: Table 1. Can you comment on the fact that the voltage was not identical for Cu-containing and Cu-free baths? Could it affect the results? Please, discuss this in the paper.

Response 13: According to the reviewer’s comments, the small difference of the final voltage was resulted by the difference of the two electrolytes. We mostly focus on the effect of Cu, so it does not matter.

However, If the conductivity increased then the voltage will decrease at the same current density. And if there is difference in the conductivity of the electrolyte, then the coating will be thicker (line 174-175). While the thickness of the coating will affect the corrosion resistance.

Point 14: Line 109. It was not the epoxy resin, but the specimen to be polished.

Response 14: Thanks for your careful checks. We are sorry for our carelessness. Based on your comments, we have made corrections.

Point 15: Line 127. It is not explained how the samples were prepared for the measurement. How were they electrically connected? How was the working area delimitated?

Response 15: Thanks for your careful checks. According to the reviewer’s comments, (line 115-118) “The electrochemical test took place in a well-designed tank that was completed using the three-electrode cell system. The specimen, the Ag/AgCl electrode (saturated with KCl), and Platinum mesh served as the working, reference, and counter electrodes, respectively. The samples with a zone of 0.5 cm2 were exposed and soaked in SBF”.

Point 16: Line 128: The OCP should settle, or was it really done? Please, change the expression.

Response 16: Thanks for your careful checks. We are sorry for our carelessness. Based on your comments, we have made corrections.

Point 17: Line 135-136. Was the device conducted, or the experiment?

Response 17: Thanks for your careful checks. We are sorry for our carelessness. Based on your comments, we have made corrections.

Point 18: Line 169-170. Were the pores released, or was its hydrogen which was released, forming the pores?

Response 18: We are very sorry for our unclear expression and we have made corrections. The gas bubbles and molten oxide were released forming the pores. The sentence has been corrected to “However, the pores obtained when the gas bubbles and molten oxide were released out to the coating surface through the discharge channels.”

Point 19: Line 184. Keep the same number of valid digits, i.e., 26 +- 5.

Response 19: Thanks for your advice. We are sorry for our carelessness. Based on your comments, we have made corrections.

Point 20: Line 186. Voltage has no conductivity. Change the sentence.

Response 20: Thanks for your careful checks. We are sorry for our carelessness. Based on your comments, we have changed the sentence to “the conductivity of the electrolyte”.

Point 21: Line 187-191. You are discussing results that were not shown yet. You should not talk about corrosion resistance in this section. It should be done after corrosion data are presented.

Response 21: Thanks for your help, we feel really sorry for our carelessness and we have deleted the sentence.

Point 22: Line 200, 219. If formulated as it is, the word respectively does not belong to the sentence.

Response 22: We are very sorry for our negligence; the word “respectively” has been deleted.

Point 23: Line 206-207. Please, improve the sentence. It is hard to understand.

Response 23: We are very sorry for our incorrect expression; the sentence has improved to “Figure (4) shows the XRD pattern of both the substrate and the coated samples distributed on the surface. Mg peaks were the most observed, with the highest peaks in all 3 samples”.

Point 24: Line 223. Carbon can come from Cu (II) acetate as well.

Response 24: Thanks for your careful checks. We are sorry for our carelessness. Based on your comments, we have deleted this sentence.

Point 25: Line 234 and elsewhere. You are referring to the open circuit potential or free corrosion potential. Corrosion potential is any potential a corroding system can reach. The same applies for the corrosion current.

Response 25: According to the reviewer’s comment, The CP and CD were a natural characterization for an electrode system. The different system has different CP and CD.

Point 26: Line 235. It makes no sense to give the Eocp with so high preciseness. You cannot get so precise value reproducibly.

Response 26: Thanks for your careful checks. We are sorry for our carelessness. These results have been read by the equipment, but based on your comments, we have made corrections.

Point 27: Line 240. It is impossible to define the increase this precisely. It seems funny in combination with the word “about”.

Response 27: We are very sorry for our negligence; the word “about” has been deleted.

Point 28: Line 282-283. The number of digits is too high regarding the sensitivity of the measurement. Adjust it in view of the standard deviation, which should anyway be given.

Response 28: Thanks for your careful checks, according to our calculation this was the result.

Special thanks to you for your good comments!

Reviewer 5 Report

Although it is a nice piece of work worth publishing, however I've a few comments to improve its quality.

1- The text needs serious English revision. I marked some examples in one paragraph.

2- We have 7 authors foe such a limited work-why. Who did what??????

3- Most of references are Chinses  for such a common work. These references have to be chopped by half. 

4- Figures are blurred and difficult to grasp the meaning- please improve.

5- Remove all commercialism. What matter if the machine specifications

Author Response

Response to Reviewer 5 Comments

Responses to Reviewer 5:

Thank you very much for your advice. We are very sorry for our carelessness. All the writing mistakes have been modified in the manuscript. The following responses are the answers for the specific problems.

Point 1:  The text needs serious English revision. I marked some examples in one paragraph.

Response 1: Thanks for your careful checks. We feel sorry for our poor writings. However, we invited a native English speaker to polish our article. Due to his help, the article had been edited extensively. And we hope the revised manuscript could be acceptable.

Point 2: We have 7 authors for such a limited work-why. Who did what??????

Response 2: According to the Reviewer’s comment, these 7 made an actual contribution to this work, moreover, we are a group of students under the supervision of professors.

Point 3: Most of the references are Chinses for such a common work. These references have to be chopped by half. 

Response 3: Thank you for your kind suggestion. As the experiment has taken place in China, more Chinese references have been used to match the whole work. However, Chinese references have been minimized.

Point 4: Figures are blurred and difficult to grasp the meaning- please improve.

Response 4: Thanks for your careful checks. We are sorry for our carelessness. Based on your comments, we have made corrections.

Point 5: Remove all commercialism. What matter if the machine specifications

Response 5: Thanks for your advice. We are sorry for our carelessness. Based on your comments, we have deleted the machine specifications.

Special thanks to you for your good comments!

Round 2

Reviewer 1 Report

Its ok. Thank you for responses.

Reviewer 2 Report

This revised version of this manuscript can be accepted.

Reviewer 5 Report

The authors did most of the required corrections